# Rosmarinic Acid, the Main Effective Constituent of Orthosiphon stamineus, Inhibits Intestinal Epithelial Apoptosis Via Regulation of the Nrf2 Pathway in Mice

**DOI:** 10.3390/molecules24173027

**Published:** 2019-08-21

**Authors:** Xuan Cai, Fan Yang, Lihui Zhu, Ye Xia, Qingyuan Wu, Huiqin Xue, Yonghong Lu

**Affiliations:** 1Shanghai Shenfeng Animal Husbandry and Veterinary Science Technology Co., Ltd., Shanghai 201106, China; 2Institute of Animal Husbandry & Veterinary Science, Shanghai Academy of Agricultural Science, Shanghai 201106, China; 3Shanghai Engineering Research Center of Breeding Pig, Shanghai 201106, China; 4Biology Department, College of Life and Environment Science, Shanghai Normal University,100 Guilin Road, Shanghai 200234, China; 5State Key Laboratory of Bioreactor Engineering, East China University of Science and Technology, 130 Meilong Road, Shanghai 200237, China; 6Shanghai Collaborative Innovation Center for Biomanufacturing Technology, 130 Meilong Road, Shanghai 200237, China

**Keywords:** antioxidants, apoptosis, orthosiphon stamineus, rosmarinic acid, Nrf2

## Abstract

Many studies have shown that Orthosiphon stamineus extract (OE) has antioxidant activity, and we previously reported that OE protects the intestine against injury from a high-fat diet. However, the molecular mechanism underlying this protective effect of OE was unclear. Here, OE was separated according to polarity and molecular weight, and the antioxidant activity of each component was compared. The components with the highest antioxidant activity were analyzed by HPLC, which confirmed that rosmarinic acid (RA) was the main effective constituent in OE. OE and RA were then tested in a mouse high-fat diet-induced intestinal injury model. The antioxidant indices and morphological characteristics of the mouse jejunum were measured, and activation of the nuclear factor E2-related factor 2 (Nrf2) pathway and apoptosis of jejunal epithelial cells were analyzed. Of all the constituents in OE, RA contributed the most. Both RA and OE activated the Nrf2 pathway and increased downstream antioxidant enzyme activity. RA and OE protected the mouse intestine against high-fat diet-induced oxidative stress by preventing intestinal epithelial cell apoptosis via both extracellular and intracellular pathways. Thus, RA, the main effective constituent in OE, inhibits intestinal epithelial apoptosis by regulating the Nrf2 pathway in mice.

## 1. Introduction

Owing to rapid economic development, dietary habits in China and other eastern countries are now similar to the “Western diet,” including excess energy from fat. However, a high-fat diet causes oxidative stress in the intestines [1].

*Orthosiphon stamineus* Benth. (*Clerodendranthus spicatus* Thunb.), more commonly known as “java tea,” is widely grown throughout Southeast Asia, Australia, and southern China. It has been consumed as tea for thousands of years in south Asia, and in China, it is usually used as a traditional medicinal herb to treat kidney stones [2]. Recent research has shown that Orthosiphon stamineus extract (OE) has anti-angiogenic, anti-tumor, anti-inflammatory, anti-hyperglycemic, and anti-hypertensive properties [3]. The antioxidant activity of OE has been extensively studied. OE contains three main types of phytochemicals: flavonoids, caffeic acid derivatives, and terpenoids [3]. Rosmarinic acid (RA) is the main phenolic constituent in the alcohol (methanol)-water extract [4], and RA has been reported to have an anti-apoptotic effect in oxidative stress-induced apoptosis [5]. 

We previously showed that OE protected the intestine against oxidative stress [6]. However, there are no reports confirming that RA is the predominant effective compound in the antioxidant activity of OE. In addition, the molecular mechanism underlying the scavenging of reactive oxygen species (ROS) by OE is unknown. 

Oxidative stress causes apoptosis in animals [7]. The nuclear factor E2-related factor 2 (Nrf2) pathway is regarded as the major mechanism protecting against oxidative stress because it modulates the expression of hundreds of genes, including those encoding antioxidant enzymes and their relative responses [8]. Therefore, we hypothesized that OE might protect the intestinal epithelium against apoptosis via the Nrf2 pathway. In this study, we confirmed that RA is the predominant effective compound in OE. A high-fat diet-induced oxidative stress mouse model was used to examine whether OE regulates intestinal cell apoptosis and to assess the influence of OE on the Nrf2 pathway and investigate the underlying mechanism.

## 2. Results

### 2.1. RA is the Main Antioxidant in OE

The composition of OE was determined and is shown in Figure 1a. OE was found to contain 12.05% polysaccharide and 7.80% protein. The content of total polyphenol and total flavonoids in OE was 192 μg/mg gallic acid (GA) equivalent and 352 μg/mg rutin equivalents, respectively. Structures of the main compounds in OE are shown in Figure 1b. Reducing power reflects the ferric ion reducing antioxidant power; 1,1-diphenyl-2-picrylhydrazyl (DPPH) is a nitrogen radical, and superoxide anion is an oxygen radical. Figure 1c shows that water phase with high molecular weight (HE) had lower antioxidant activity than OE, thus indicating that macromolecular compounds are not the main antioxidant compounds in OE. Thus, small molecule content (LE) was subsequently analyzed. LE was separated by polar liquid–liquid extraction. After at least three liquid–liquid extractions, most LE (75.68%) was in the ethyl acetate phase, and some were lost in the extraction (others, 6.33%). Compared with the water extract (WE) and petroleum ether extract (PE), the ethyl acetate extract (EE) showed much higher anti-oxidation capacity. Our previous study [6] confirmed that the phenolic acids and flavonoid compounds were sinensetin, eupatorin, 3’-hydroxy-5,6,7,4’-tetramethoxyflavone (TMF), RA, and caffeic acid, as confirmed by the present study (Appendix A). The antioxidant activity of these compounds was also determined (Figure 1d). RA not only had the highest content in EE but also had the highest antioxidant activity among the compounds. These results indicate that RA is the main contributor to the antioxidant activity of OE.

### 2.2. OE and RA Showed Antioxidant Activity in vivo But Did Not Alter Body Weight

The high-fat diet caused an increase in body weight during the 8-week feeding trial (normal diet (NC), 26.34 ± 0.46 g; high-fat diet (FC), 33.64 ± 0.34 g; or high-fat diet with OE, 33.09 ± 0.36 g or RA, 33.89 ± 0.40 g), but OE and RA did not alter body weight compared with the mice in FC group (Appendix A, *P* > 0.05).

The depletion of superoxide dismutase (SOD), catalase (CAT), and glutathione peroxidase (GSH-Px) activity in an organism indirectly reflects the generation of ROS. Malondialdehyde (MDA) is a reliable biomarker of lipid peroxidation, and 8-OH-dG is a major product of DNA oxidation. The activity of SOD, CAT and GSH-Px is shown in Figure 2e, and the content of glutathione (GSH), GSH/GSSG, MDA, and 8-hydroxy-2-deoxyguanosine (8-OHdG) is shown in Table 1. Mice fed a high-fat diet for 8 weeks showed a significant decline in SOD and GSH-Px activity (*P* < 0.05). OE significantly increased GSH-Px activity, and RA significantly increased SOD activity in the jejunum, as compared with that in the FC group (*P* < 0.05). MDA and 8-OH-dG concentrations increased significantly in the jejunum in mice fed a high-fat diet, and both OE and RA decreased the concentrations of MDA and 8-OH-dG. No difference was observed between the OE and RA groups in oxidative stress related parameters (*P* > 0.1).

### 2.3. OE and RA Antioxidant Activity via the Nrf2 Pathway

Although OE and RA have been confirmed to scavenge free radicals both in vitro and in vivo, the mechanism remains unclear. The Nrf2 pathway is a classic antioxidant pathway. This study showed that both OE and RA increased Nrf2 mRNA transcription (Figure 2a), but the results were not significant (*P* > 0.05). Figure 2b shows that OE and RA increased Nrf2 accumulation in the nuclear fraction. To evaluate the role of Nrf2 in the transcriptional activation of antioxidant-response elements (AREs), we performed electrophoretic mobility-shift assay (EMSA) analysis. OE and RA resulted in enhanced (Figure 2c) ARE-binding activity of Nrf2 (*P* < 0.05, compared with that in the FC group). Binding competition experiments with an ARE cold probe and a cold probe with the core of the ARE sequence mutated confirmed the specificity of the complex formation (Figure 2c). These findings indicated that OE and RA activated the Nrf2 pathway mainly by increasing Nrf2’s release from kelch-like ECH (Epichlorohydrin) associated protein 1 (Keap1) and its transport into the nucleus but did not increase mRNA transcription of Nrf2. Heme oxygenase-1 (HO-1), NAD(P)H quinone oxidoreductase 1 (NQO1), glutathione S-transferase (GST), and SOD are downstream genes that are regulated by AREs. Figure 2d shows that OE and RA significantly up-regulated GST mRNA transcription (*P* < 0.05), and OE altered SOD mRNA transcription (*P* < 0.1), as compared with that in the FC group, but not HO-1 and NQO1. The activity of SOD, CAT and GSH-Px was also assessed; OE significantly promoted GSH-Px activity (Figure 2e), whereas RA significantly promoted SOD activity (*P* < 0.05). These results further suggested that Nrf2 mediates OE and RA-induced activation of antioxidant activity in the mouse jejunum.

### 2.4. OE and RA Alter the Ratio of Villus and Crypt Structure in Mice

Mice fed a normal diet had intact, long, and integrated villi (Figure 3a), whereas mice fed a high-fat diet (Figure 3b) showed marked shortening, clubbing, and blunting of villi. The villi of mice treated with OE (Figure 3c) or RA (Figure 3d) exhibited morphological changes, as compared with those in the control mice fed a normal diet, but these changes were different from those in mice fed a high-fat diet and treated with PBS. These changes include shorter villi and increased crypt depth [9]. Figure 3e–g shows that both OE and RA protected intestinal villi from acquiring shorter villi and increased crypt depth as a result of the high-fat diet.

### 2.5. OE and RA Inhibit Apoptosis of Intestinal Cells

Figure 4a shows an elevated number of apoptotic cells (apoptotic nuclei are stained dark brown) in the intestinal villi in high-fat diet fed mice. Moreover, the number of apoptotic cells per villus was higher (*P* < 0.05) in high-fat fed mice (FC, OE, and RA groups) than in control mice (NC group). Fewer (*P* < 0.05) apoptotic cells were present at the villus in the OE and RA groups than the FC group. Figure 4b shows that the expression of cleaved caspase-3, -8, -9, cytochrome C, and Fas ligand was substantially lower in the RA group than the FC group. OE also significantly decreased levels these apoptosis-associated proteins, except for caspase-9. No differences between OE and RA treated mice were observed (*P* > 0.1). To further confirm the anti-apoptotic effect of OE and RA in the mouse jejunum, we analyzed the activity of caspase-3, -8, and -9 (Figure 4c). High-fat diet feeding for 8 weeks significantly increased the activity of caspase-3, -8, and -9. However, the activity of caspase-3, -8, and -9 in mice treated with OE and RA at doses of 100 mg/kg and 2 mg/kg body weight, respectively, returned to normal levels (*P* > 0.1 compared with the NC group). These findings indicated that the high-fat diet-induced increase in apoptosis can be reversed by OE or RA, and the anti-apoptotic effect of OE and RA is mediated by Fas and mitochondria.

## 3. Discussion

Many studies have shown that OE has a very high total phenol content and antioxidant activity, and RA, TMF, eupatorin, sinensetin, and caffeic acid are the major antioxidant components [3,10], as confirmed in our previous study [6]. However, no systematic research has been performed to confirm the major component of the antioxidant activity of OE. A study by Matkowski et al. [11] has shown that hydrophilic compounds play a major role in determining antioxidant capacity; therefore, the components with a molecular weight greater than 1000 Da were separated in the present study, and the lipid soluble constituent was extracted with petroleum ether. The hydrophilic compounds with low molecular weight (EE) had the highest antioxidant activity in OE. RA had not only the highest content of polyphenols but also the highest antioxidant activity among all the major small compounds in OE. Thus, RA was selected to determine the mechanism of OE in high-fat diet-treated mice. The concentration of RA used in the animal experiments was the same as that in OE (approximately 2%). Interestingly, EE showed similar (but lower than RA) antioxidant activity to RA, although only 19.65% of RA was found in EE. Other compounds with high antioxidant activity may be present in OE, or, more likely, these polyphenols may interact [12].

The intestinal epithelium is a highly proliferative and apoptotic tissue. Intestinal epithelial cells originate from proliferation of crypts (intestinal stem cells); the precursors then migrate toward the tip of the villus and differentiate into various functional cells, and the villus tip cells die from apoptosis and are shed into the lumen. Oxidative stress-induced cell apoptosis [13] and high-fat diet-induced oxidative stress have been confirmed in many studies [14]. Therefore, a high-fat diet may injure animal mucosal integrity by altering the luminal redox status [15]. This result was confirmed by hematoxylin and eosin (H&E) histology (Figure 3); high-fat diet induced apoptosis was observed in the intestinal epithelium and subsequently confirmed by TUNEL and caspase-3 activity analyses. These findings were consistent with those of Wang et al. in rats [16]. 

Apoptotic pathways are usually classified into extrinsic and intrinsic pathways [17,18]. Fas and FasL mediate the extrinsic pathway, and caspase-8 is the key downstream gene [18]. The intrinsic pathway is also called mitochondrial-mediated apoptosis. In this pathway, apoptotic stimuli exert their effects on mitochondria, thus resulting in the release of cytochrome C, and then induce caspase-9 [18]. Caspase-3 is activated by both caspase-8 and caspase-9, and it is considered a biomarker of apoptosis. This study showed that both OE and RA arrested apoptosis induced by a high-fat diet. This result was consistent with the findings of Gao et al. [19] and Domitrović et al. [20]. However, other studies have shown that RA also arrests the cell cycle during neovascularization [21] and inhibits the proliferation of colon cancer cells [22]. Thus, RA shows organ- and species-specific activity [23]. The exact mechanism is worthy of further examination.

Among the many factors that induce cell apoptosis, ROS are known to be one of the most important [13]. Studies have shown that a high-fat diet induces oxidative stress in animals [1], as confirmed in the present study by measurement of jejunal antioxidant parameters. RA has been shown to have good antioxidant activity [24], as does OE [3,6]. Our in vitro and in vivo research additionally confirmed these results. Therefore, we suggest that OE and RA regulate jejunal epithelium by scavenging ROS and activating an anti-apoptotic mechanism. 

Under normal conditions, Nrf2 is sequestered by Keap1 in the cytosol. When the oxidative stress sensor Keap1 encounters ROS or electrophilic chemicals, Nrf2 is released, translocates into the nucleus and binds to the ARE, thus stimulating the expression of GSH, HO-1, NQO1, GST, and many other oxidized protein repair genes. Nrf2 also up-regulates the anti-apoptotic protein Bcl-2 and is involved in cross-talk with other pathways, such as that of NF-κB [25]. This study showed that OE and RA increased Nrf2 protein in the nucleus and increased ARE element binding; the transcription of downstream proteins also significantly increased. These findings indicate that OE and RA act against oxidative stress via the Nrf2 pathway, and the anti-apoptotic mechanism of OE and RA in mouse jejunal epithelium may be related to the Nrf2 pathway.

A high-fat diet leads to obesity, diabetes, cardiovascular disease, and even learning disorders [1]. A high-fat diet is digested and absorbed in the gastrointestinal tract and is known to result in altered intestinal microbiota and inflammatory bowel disease [26]. Protection against intestinal injury due to a high-fat diet can prevent the health risks associated with the Western diet. In our study, although the body weight of mice treated with OE or RA did not significantly decrease (Appendix A), the blood lipid levels in both the OE and RA group decreased (*P* < 0.05). Other physiological parameters in the OE or RA group, such as glutamic-pyruvic and transaminase activity, were also better than those in the FC group. These findings indicate that OE may be a good dietary supplement for those who prefer high-fat or high energy food.

## 4. Materials and Methods 

Please refer to Appendix A for the details of the materials and methods.

### 4.1. Plant Materials

*O. stamineus* was collected in Yulin (Guangxi province, China) in June, 2017. The plant was identified by Dr. H. B. Hu (Key Laboratory of Natural Drug Research and Development, Gannan Medical University), and a voucher specimen was retained in our laboratory for future reference (Voucher Number: OSL2017.06YL-002).

### 4.2. Solvent Extraction and Partition Fractionation

*O. stamineus* was dried under direct sunlight [27]. Dry powdered leaves (1000 g) were extracted with 50% aqueous ethanol with an ultrasonic extractor at 50 °C for 30 min. Figure 5 shows the flow chart for the preparation of the ethanol extract and three fractions (petroleum ether phase, PE; water phase with high molecular weight, HE; water phase with low molecular weight, LE) of *O. stamineus*. LE was then analyzed by HPLC to determine the major antioxidant compounds.

### 4.3. Composition of OE

Protein in OE was measured by the Coomassie brilliant blue method. Polysaccharide was analyzed with the phenol-sulfate method. The total polyphenol content in the extract was determined with the Folin–Ciocalteu method using gallic acid (GA) as the standard. Total flavonoids in the extract were determined with rutin as the standard. The major antioxidant compounds in OE were detected by HPLC using an ultraviolet detector.

### 4.4. Reducing Power and Free Radical Scavenging Activity Analysis

The reducing power and DPPH radical and superoxide anion scavenging activity of OE and the main antioxidants were determined with assay kits, according to the manufacturer’s instructions (Congyi Bio Technology, Shanghai, China). The reducing power was expressed as absorbance detected. DPPH and superoxide anion radical scavenging activity were calculated as a ratio, and they were set as 0% in a negative control (PBS)

### 4.5. Animals and Treatment

Forty male BALB/c mice weighing 18–20 g were purchased from Shanghai SLAC Laboratory Animal Co., Ltd. After acclimation for one week, the mice were randomly divided into the NC, FC, OE, and RA groups, with ten mice per group. The NC group received only a normal diet (D12450B, Research Diet Inc., New Brunswick, U.S.) containing 4.3% fat, and the other groups received a high-fat diet (D12492, Research Diet Inc.) containing 35% fat. The mice in the OE and RA groups were orally administered OE or RA at doses of 100 mg/kg and 2 mg/kg body weight, respectively, and the mice in the NC and FC groups were orally administered saline for eight weeks. At the end of the study, blood samples were collected after eyeball removal, and jejunal samples were washed immediately with ice-cold PBS and stored at −80°C before analysis. These experiments were carried out in accordance with local guidelines for the care of laboratory animals and were approved by the institution’s ethics committee for research using laboratory animals, approval code: SN-XS-20170048.

### 4.6. Measurement of Oxidative Stress in Mice

Oxidative stress was assessed by quantitative measurement of hydrogen peroxide, GSH, GSH/GSSG (glutathione disulfide), MDA, 8-OHdG, GSH-Px, CAT, GST, and SOD activity. The 8-OHdG and 8-iso-PG were detected with an ELISA kit purchased from Cusabio (Wuhan, China) and Shanghai Enzyme-linked Biotechnology (Shanghai, China), respectively. Other parameters were detected with assay kits according to the manufacturer’s instructions (Nanjing Jiancheng Bioengineering Institute, Nanjing, China).

### 4.7. Quantitative Real-Time PCR Analysis

Total mRNA was extracted from intestinal epithelial tissue with TransZol Up reagent (Transgen Biotech, Beijing, China) and immediately reverse transcribed to cDNA according to the manufacturer’s instructions (Takara, Japan). The quality of total RNA was assessed. A real-time PCR assay was used to quantify gene expression. The primer sequences and reaction system are shown in Appendix A. The relative expression of mRNA species was determined with the comparative Ct method [28]. 

### 4.8. EMSA

Nuclear Nrf2 DNA binding activity was measured with an EMSA kit (Thermo Fisher), according to the manufacturer’s instructions. Oligonucleotides corresponded to the binding sites of the ARE nucleotide. The oligonucleotide sequences are shown in Appendix A. The specificity of the band was confirmed by competition with cold oligonucleotides and mutated oligonucleotides. Gels were imaged with a scanner (V300, Epson, Japan) and then analyzed with the alphaEaseFC system (Alpha Innotech, Portugal).

### 4.9. Measurement of Villus Height and Crypt Depth

Jejunal tissues were collected and processed according to standard protocol. Briefly, the tissues were fixed in 4% neutral buffered formalin, embedded in paraffin wax, cut into 5–10 µm sections with a rotary microtome and stained with H&E. Villus height was calculated from the base of the villus, above the level of adjoining crypts, to the villus tip. The depth of crypts was calculated from the boundaries between the crypts and myenteron to the base of the villus. At least ten typical villi and crypts were measured for each animal, and the mean value was calculated.

### 4.10. DNA Fragmentation Analysis

A TUNEL kit (Boster, Wuhan, China) was used for DNA fragmentation analyses. All operations were conducted according to the manufacturer’s protocol. For each slide, five fields were randomly chosen from each section under the microscope. The apoptotic index was determined as the number of apoptotic cells per villus.

### 4.11. Analysis of Caspase-3, -8, and -9 Activity

The activity of caspase-3, -8, and -9 was measured with substrate peptides Ac-DEVD-pNA, Ac-IETD-pNA, and Ac-LEHD-pNA with caspase activity kits (Beyotime, Shanghai, China). 

### 4.12. Western Blot Analysis

Jejunal tissue was analyzed in this study. For Nrf2 determination, the nuclear extracts were prepared with a nuclear protein extraction kit (Beyotime, Shanghai, China). Samples were electrophoresed with 10% SDS-PAGE, transferred onto polyvinylidene fluoride membranes and then incubated with specific primary antibodies. The membranes were washed and then incubated with horseradish peroxidase–linked secondary antibody. The density of the bands was quantified with enhanced chemiluminescence (ECL) reagent and analyzed with a gel analysis system. The antibody catalog number is shown in Appendix A.

### 4.13. Statistical Analysis

All reaction mixtures were prepared in triplicate, and at least three independent assays were performed for each sample. All data are expressed as mean ± SEM. Data were analyzed by one-way ANOVA followed by Duncan’s multiple range tests in SPSS version 17.0 software. A P value < 0.05 was considered statistically significant.

## 5. Conclusions

In conclusion, this study confirmed that RA is the most effective component responsible for the antioxidant activity of OE. RA and OE protected mouse intestines against high-fat diet-induced oxidative stress by preventing intestinal epithelial cell apoptosis. Both RA and OE activated the Nrf2 pathway and increased downstream antioxidant enzyme activity. Because OE has been demonstrated to be safe, we suggest that it may be a good dietary supplement for those who prefer high-fat or high energy food.

## Figures and Tables

**Figure 1 molecules-24-03027-f001:**
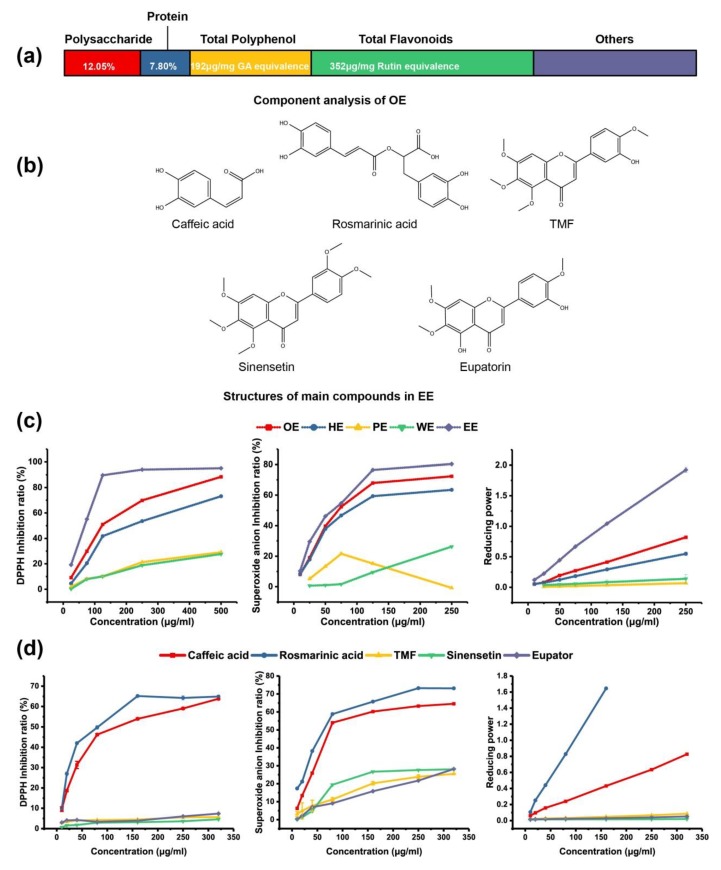
Content and antioxidant activity of the different components in Orthosiphon stamineus extract (OE). (**a**) Component analysis of OE. (**b**) Structures of the main compounds in ethyl acetate extract (EE). (**c**) Antioxidant activity of OE, water phase with high molecular weight (HE), water phase with small molecule content (LE), petroleum ether extract (PE), water extract (WE), and EE. (**d**) Antioxidant activity of the main compounds in EE. TMF = 3’-hydroxy-5,6,7,4’-tetramethoxyflavone (TMF).

**Figure 2 molecules-24-03027-f002:**
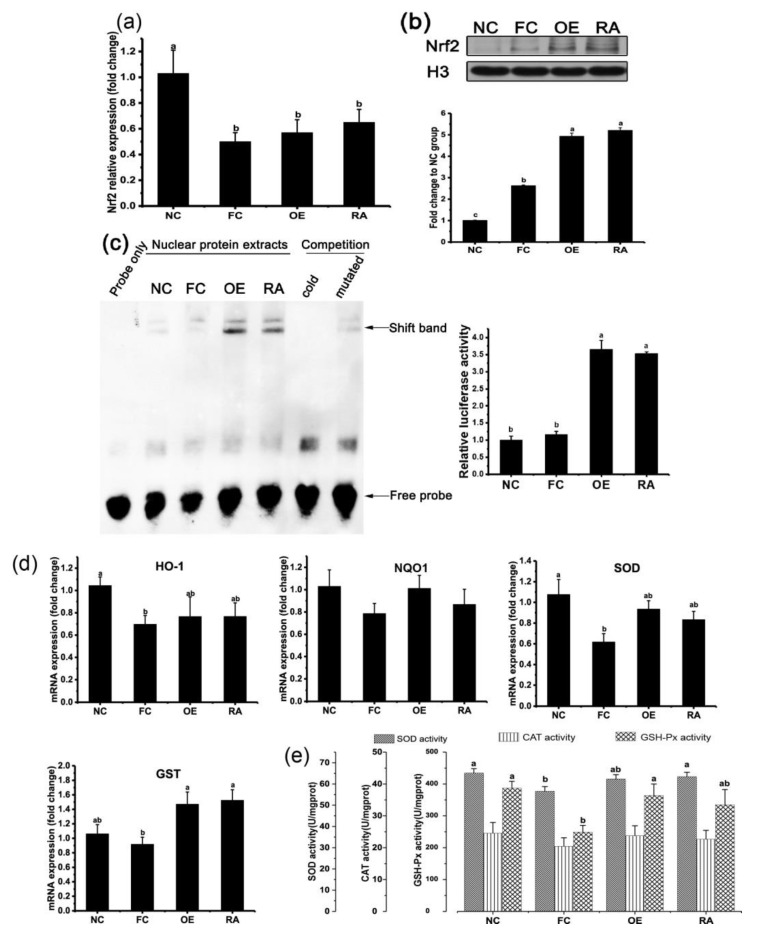
OE and RA activated the nuclear factor E2-related factor 2 (Nrf2) pathway. (**a**) Nrf2 mRNA transcription in the mouse jejunum after different treatments. (**b**) OE and RA trigger nuclear translocation of Nrf2 in the mouse jejunum. (**c**) Nrf2 binding to specific antioxidant-response element (ARE) sites, determined by electrophoretic mobility-shift assay (EMSA). The probe only lane contains a labeled probe with no protein, and nuclear protein extract lanes contain the probe with mouse jejunal epithelial cell nuclear protein extracts. Competition lanes contain 200× unlabeled probe. Gels were read with a scanner, and optical density is expressed as fold change in the NC group. (**d**) Effects of OE and RA on mRNA expression of downstream antioxidant related genes. (**e**) Activity of superoxide dismutase (SOD), catalase (CAT), and glutathione peroxidase (GSH-Px) in the mouse jejunum. The bar represents the SEM, with different letters being statistically significantly different in all figures (*P* < 0.05).

**Figure 3 molecules-24-03027-f003:**
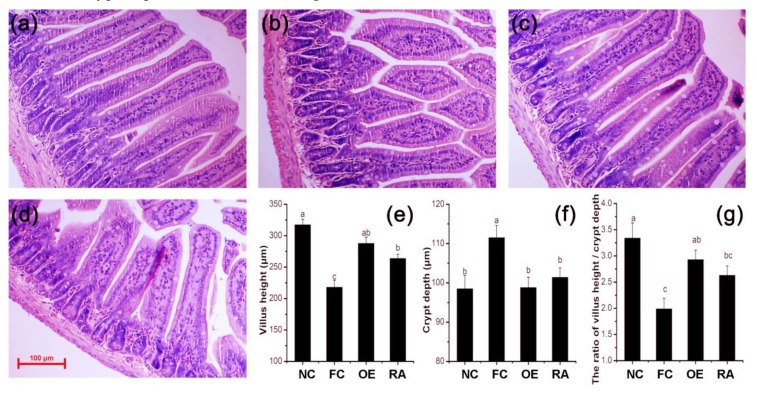
OE and RA alter jejunum villus and crypt structure in mice. (**a**–**d**) Representative hematoxylin and eosin (H&E) histology of the jejunum in mice after different treatments. (**a**) NC group; (**b**) FC group; (**c**) OE group; (**d**) RA group. (**e**,**f**,**g**): At least ten typical villi were measured for each mouse. The bar represents the SEM, with different letters being statistically significantly different (*P* < 0.05). A higher magnification figure is shown in Appendix A.

**Figure 4 molecules-24-03027-f004:**
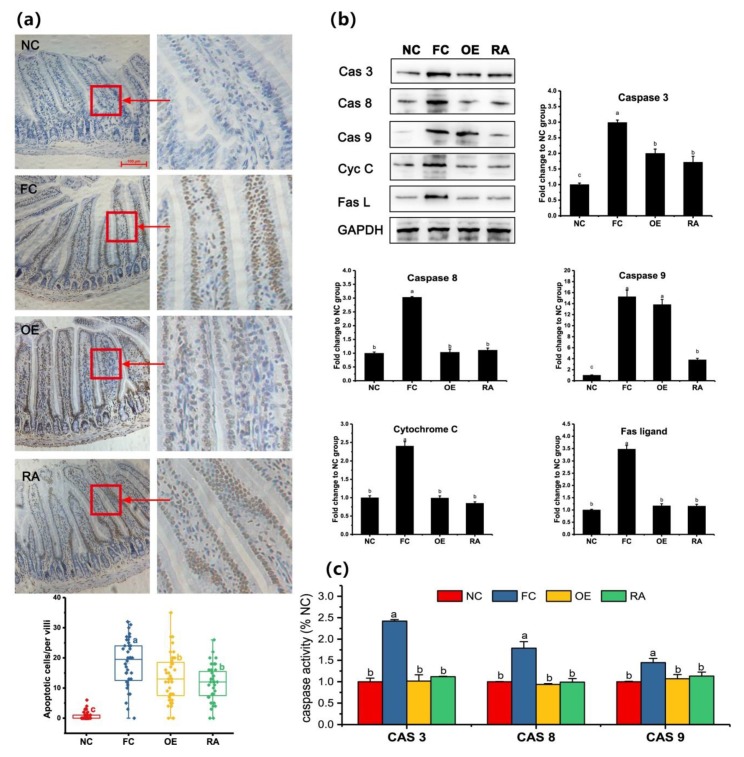
OE and RA inhibit intestinal cell apoptosis induced by a high-fat diet. (**a**) Terminal-deoxynucleoitidyl Transferase Mediated Nick End Labeling (TUNEL) assays of mouse jejunum after different treatments. The experiment was carried out with a kit from Boster (Wuhan, China). Apoptotic nuclei stain brown, and normal nuclei stain blue. The box-plot figure was draw in Origin software, and data are compared with Dunnett T3 by ANOVA (*n* = 40). (**b**) Protein lysates from jejunal segments were examined by western blotting for caspase-3, -8, and -9, cytochrome C, and Fas ligand protein levels. Reduced glyceraldehyde-phosphate dehydrogenase gene (GAPDH) was used as the control. (**c**) Caspase activity of jejunal segments after different treatments. The activity of caspases was measured with substrate peptides. For (**b**) and (**c**), data are expressed as the fold change relative to the NC group and were compared with Duncan’s test by ANOVA. Different letters indicate significant differences (*P* < 0.05). Cas 3, caspase-3; Cas 8, caspase-8; Cas 9, caspase-9; Cyc C, cytochrome C; Fas L, Fas ligand.

**Figure 5 molecules-24-03027-f005:**
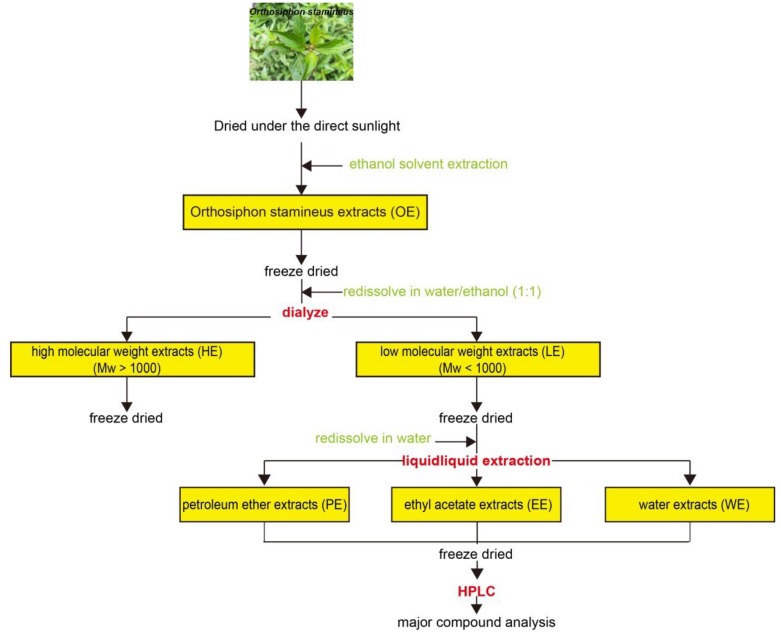
Flow chart of solvent extraction, liquid-liquid partition, and main compound analysis of *Orthosiphon stamineus*.

**Table 1 molecules-24-03027-t001:** Jejunal antioxidant parameters in mice fed a normal diet (NC), high-fat diet (FC), or high-fat diet with OE (OE) or rosmarinic acid (RA).

Items	NC	FC	OE	RA
GSH (μmol/mg prot)	5.85 ± 0.99^a^	2.70 ± 1.09^b^	4.54 ± 0.88^ab^	4.81 ± 0.81^ab^
GSH/GSSG	3.79 ± 0.23	3.16 ± 0.39	3.57 ± 0.32	3.48 ± 0.51
MDA (nmol/mg prot)	2.82 ± 0.26^c^	6.38 ± 0.66^a^	4.16 ± 0.58^bc^	4.60 ± 0.39^b^
8-OH-dG (pg/mg prot)	72.43 ± 12.94^b^	294.21 ± 27.22^a^	121.54 ± 27.91^b^	171.17 ± 42.55^b^

^ab^ Means within the same row with different superscripts differed significantly (*P* < 0.05). GSH, glutathione; GSSG, glutathione disulfide; MDA, malondialdehyde; 8-OH-dG, 8-hydroxy-2´-deoxyguanosine.

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
