# Peer review of "Rosmarinic Acid, the Main Effective Constituent of Orthosiphon stamineus, Inhibits Intestinal Epithelial Apoptosis Via Regulation of the Nrf2 Pathway in Mice"

_molecules, 2019, doi:10.3390/molecules24173027_

Round 1

Reviewer 1 Report

In this manuscript, the authors investigated the prosurvival effects of RA and OE on fat-induced intestinal cell apoptosis in animal model. They found that RA and OE suppressed epithelial cell apoptosis via mitochondrial and external pathways. This protective effect was associated with activation of Nrf2 pathway. This study is interesting and of clinical relevance to high fat diet-induced organ damage.

There are some points should be fixed before considering this paper acceptable for publication:

1-In Figures 3 and 4, the authors should add high magnification figures showing epithelial cell apoptosis.

2-Regarding 8-OHdG and Nrf2, the authors better to show figures showing the nuclear localization of these proteins in various groups using IHC or IF methods.

3-It is better to add a diagram showing the summary of results and conclusions.

Author Response

1Comments and Suggestions for Authors

In this manuscript, the authors investigated the prosurvival effects of RA and OE on fat-induced intestinal cell apoptosis in animal model. They found that RA and OE suppressed epithelial cell apoptosis via mitochondrial and external pathways. This protective effect was associated with activation of Nrf2 pathway. This study is interesting and of clinical relevance to high fat diet-induced organ damage.

There are some points should be fixed before considering this paper acceptable for publication:

1-In Figures 3 and 4, the authors should add high magnification figures showing epithelial cell apoptosis.

A: Thank you for your advice. For Figure 3, now we add high magnification figures about intestinal villus and epithelial cells in supplementary material “S4”.The photo showed in figure 3 by 100 x magnifications, and the photo showed in S4 by 200 x magnifications. We also took some 400x photos, but we didn’t show in figure 3 because higher magnification figure cannot show a comprehensive view of villus. For figure 4, we have added high magnification figures now.

2-Regarding 8-OHdG and Nrf2, the authors better to show figures showing the nuclear localization of these proteins in various groups using IHC or IF methods.

A: Thank you for your advice. This is real a good suggestion, but unfortunately, this study already finished in August, 2017. Though paraffin-embedded samples were still kept, further analysis by IHC or IF methods may be very difficult. In other word, this study analyzed nuclearNrf2 concentration by western-blot method. In a sense, it also showed localization of Nrf2.

3-It is better to add a diagram showing the summary of results and conclusions.

A: Thank you for your advice. This manuscript includes a graphical abstract, it is a summary of results and conclusions. It may be not included in the main PDF document, but it has been uploaded. For convenience reading, we paste it in follow.

Reviewer 2 Report

This manuscript describes Rosmarinic acid and Orthosiphon stamineus inhibit intestinal epithelium apoptosis via regulation of the Nrf2 pathway in mice. The in vitro and in vivo studies were both conducted. However, this manuscript has major concerns as follows, and a major revision is needed before possible consideration.

The introduction section should be concise and a simple description of Nrf2 pathway should be added. All figures and tables should be cited in the right place in the main text (e.g. line 134 and line 152). Pay attention to the manuscript style of Molecules. The footer of Table 1 is missing; the space between paragraphs (line 102-103, 132-133). An abbreviation list is needed in this manuscript. More detailed information in Figure 4 should be added such as dose… The manuscript suffers numerous language issues and should be polished by a native English speaker. Some examples with the poor expression are listed below (not a full list): Line 134 – 139 “Figure 3 shows that mice fed a normal diet had intact, long and integrated villi (Fig. 4a) and mice fed a high-fat diet (Fig. 3b) had marked shortening, clubbing, and blunting of villi. The villi of mice treated with OE (Fig. 3c) or RA (Fig. 3d) exhibited morphological changes, as compared with control mice fed a normal diet, but these changes were different to those fed a high-fat diet and treated with PBS. It has been shown that these changes include shorter villi and increased crypt depth [8].” Line 211 – 222 “RA has proven to have good antioxidant activity [23] as does OE”

Author Response

2 Comments and Suggestions for Authors

This manuscript describes Rosmarinic acid and Orthosiphon stamineus inhibit intestinal epithelium apoptosis via regulation of the Nrf2 pathway in mice. The in vitro and in vivo studies were both conducted. However, this manuscript has major concerns as follows, and a major revision is needed before possible consideration.

1) The introduction section should be concise and a simple description of Nrf2 pathway should be added.

A: Thank you. This is real a good suggestion, and the introduction section has added a simple description of Nrf2 pathway (line57-59).

2) All figures and tables should be cited in the right place in the main text (e.g. line 134 and line 152).

A: Thank you. The figures and tables have been cited.

3) Pay attention to the manuscript style of Molecules.

A: Thank you. We have checked the style again.

4) The footer of Table 1 is missing; the space between paragraphs (line 102-103, 132-133).

A: Thank you. The footer of Table 1 was added already, and the space between paragraphs has been corrected.

5) An abbreviation list is needed in this manuscript.

A:Thank you. We have checked all again. However, we noticed that “Molecules” doesn’t need abbreviation list, while labeled at the first time appearance instead.

6) More detailed information in Figure 4 should be added such as dose…

A: Thank you. Figure 4 has been redrawn and a more detailed Figure 4 legend has been added.

7) The manuscript suffers numerous language issues and should be polished by a native English speaker.

Some examples with the poor expression are listed below (not a full list):

Line 134 – 139 “Figure 3 shows that mice fed a normal diet had intact, long and integrated villi (Fig. 4a) and mice fed a high-fat diet (Fig. 3b) had marked shortening, clubbing, and blunting of villi. The villi of mice treated with OE (Fig. 3c) or RA (Fig. 3d) exhibited morphological changes, as compared with control mice fed a normal diet, but these changes were different to those fed a high-fat diet and treated with PBS. It has been shown that these changes include shorter villi and increased crypt depth [8].”

Line 211 – 222 “RA has proven to have good antioxidant activity [23] as does OE” 

A: The language of this manuscript (not include this letter) is re-edited by a company named “International Science Editing”. The website of this company is “https://www.internationalscienceediting.com/”  I’m not very sure if the language of the manuscript meets the standard required for publication in Molecules, if the language of this manuscript still exists many problems, please let me know and we will choose another language edit company.

Round 2

Reviewer 2 Report

The authors have addressed all questions. I would like to suggest the acceptance of this paper for publication in Molecules.